# Expression of immunoglobulin constant domain genes in neurons of the mouse central nervous system

Louis Scheurer[1,*], Rebecca R Das Gupta[1,2,*] , Annika Saebisch[1], Thomas Grampp[1], Dietmar Benke[1],
Hanns Ulrich Zeilhofer[1,2] , Hendrik Wildner[1]

General consensus states that immunoglobulins are exclusively expressed by B lymphocytes to form the first line of defense against common pathogens. Here, we provide compelling evidence for the expression of two heavy chain immunoglobulin genes in subpopulations of neurons in the mouse brain and spinal cord. RNA isolated from excitatory and inhibitory neurons through ribosome affinity purification revealed *Ighg3* and *Ighm* transcripts encoding for the constant (Fc), but not the variable regions of IgG3 and IgM. Because, in the absence of the variable immunoglobulin regions, these transcripts lack the canonical transcription initiation site used in lymphocytes, we screened for alternative 5′ transcription start sites and identified a novel 5′ exon adjacent to a proposed promoter element. Immunohistochemical, Western blot, and in silico analyses strongly support that these neuronal transcripts are translated into proteins containing four Immunoglobulin domains. Our data thus demonstrate the expression of two Fc-encoding genes *Ighg3* and *Ighm* in spinal and supraspinal neurons of the murine CNS and suggest a hitherto unknown function of the encoded proteins.

## Introduction

Igs are heterodimeric proteins consisting of two heavy and two light chains. They are produced by B lymphocytes and exert a critical function in the adaptive immune response by specifically binding to pathogens such as bacteria and viruses aiding their neutralization. Igs are composed of two functional parts, an N-terminal variable part required for antigen binding and a C-terminal constant part for effector function (August, 2006; Schroeder & Cavacini, 2010). The variable domain of the heavy chain is encoded by $V_H$, $D_H$, and $J_H$ gene segments. It is generated through a series of complex gene rearrangements between different $V_H$, $D_H$, and $J_H$ segments and is specific for an individual B lymphocyte. Located downstream of the

$V_H$, $D_H$, and $J_H$ DNA segments are genes encoding different constant domains (Fc) (in mice: *Cμ/Ighm*, *Cδ/Ighd*, *Cγ3/Ighg3*, *Cγ1/Ighg1*, *Cγ2b/Ighg2b*, *Cγ2c/Ighg2c*, *Cε/Ighe*, and *Cα/Igha*). Each constant gene segment can generate two alternative transcripts encoding either a membrane bound or a secreted Ig form. Upon maturation, B lymphocytes express first monomeric IgM and IgD isotypes using the constant genes *Ighm* or *Ighd*, respectively. These naïve B lymphocytes are released from the bone marrow into the blood to populate the periphery via the blood and the lymphatic system (Randolph et al, 2017).

The central nervous system (CNS) possesses no lymphatic system and the blood brain barrier mostly excludes Igs from entering the healthy CNS. It has thus been termed as an immune privileged organ. Yet, several reports described the presence of Igs in neurons (Naegele et al, 1991; Dunn et al, 1995; Upender et al, 1997; Yoshimi et al, 2002; Hazama et al, 2005). It was hypothesized that this "neuronal" Igs originated from lymphocyte-derived Igs taken up by neurons.

We have recently conducted a genome wide screen for translated mRNAs in excitatory and inhibitory neurons of the spinal cord and reported the expression of the *Ighg3* gene, which encodes the Fc domain of IgG3 (Das Gupta et al, 2021). In the present report, we follow up on this initial observation and demonstrate expression of not only *Ighg3* but also of *Ighm* (encoding the Fc domain of IgM) in neurons of the spinal cord. The neuronal expression of both genes remained in Rag1 knock-out mice, which lack B and T lymphocytes. We further demonstrate that *Ighg3* is specifically expressed in a small subset of *Gad67* expressing inhibitory dorsal horn interneurons, whereas *Ighm* displayed a more widespread expression in inhibitory and excitatory neurons of the spinal cord and brain. We found no evidence for expression of $V_H$, $D_H$, or $J_H$ segments but instead provide evidence that *Ighm* transcription in neurons is initiated from a so far undetected 5′ exon. Finally, RT PCR sequencing results indicate the presence of transcripts encoding for both the membrane bound as well as the secreted form of *Ighm* and the occurrence of additional alternatively spliced variants.

[1]Institute of Pharmacology and Toxicology, University of Zurich, Zürich, Switzerland [2]Institute of Pharmaceutical Sciences, Swiss Federal Institute of Technology (ETH) Zurich, Zurich, Switzerland

Correspondence: zeilhofer@pharma.uzh.ch; hwildner@pharma.uzh.ch
*Louis Scheurer and Rebecca R Das Gupta contributed equally to this work

Our results therefore indicate the expression of the effector domain of IgM and IgG3 by neurons of the mouse CNS.

## Results

### Expression of Ig encoding gene segments in spinal neurons

We have recently isolated and compared the translatomes of inhibitory (vGAT[+] or Gad67[+]) and excitatory (vGluT2[+]) neurons of the mouse spinal cord using the Translating Ribosome Affinity Purification (TRAP) approach and subsequent RNAseq analyses (Das Gupta et al, 2021). In this comparison, we made the surprising observation that *Ighg3*, the gene that encodes the constant domain of the IgG3 immunoglobulin isotype, was expressed in inhibitory neurons of the mouse spinal cord. Here, we first confirmed this finding with in situ hybridization (ISH) experiments (Fig 1), which revealed an expression pattern highly confined to a thin layer at the border between the superficial and the deep spinal dorsal horn (Fig 1B). Subsequent in-depth analysis of the RNAseq TRAP data revealed that the only other Ig encoding gene expressed in mouse spinal cord neurons was *Ighm* (Table S1 and Fig 1). ISH confirmed its widespread expression in the dorsal and ventral spinal cord (Fig 1C). In spinal inhibitory neurons, *Ighm* was expressed at a 20–160-fold higher level than *Ighg3* (normalized read count; in Gad67[+] neurons *Ighg3* = 124 ± 21, *Ighm* = 2,611 ± 290, in vGAT[+] neurons, *Ighg3* = 20 ± 3, *Ighm* = 3,368 ± 543). These expression data further suggest that, unlike Ighg3, Ighm was not only expressed in inhibitory

but also in excitatory spinal neurons (normalized read count in vGluT2[+] neurons, *Ighm* = 2,659 ± 275) (Fig 1A and Table S1). Unlike expression of *Ighg3*, no striking enrichment of *Ighm* expression was detected in either inhibitory or excitatory spinal neurons (fold change in; *Ighg3* vGAT versus vGluT2 = 26, *Ighg3* Gad67 versus vGluT2 = 167 and *Ighm* vGAT versus vGluT2 = 1.3, Gad67 versus vGluT2 = 1).

### *Ighg3* is selectively expressed in a small subset of inhibitory spinal neurons, whereas *Ighm* displays more widespread expression throughout the CNS

Presence of Ig proteins in neurons has previously been reported (Naegele et al, 1991; Dunn et al, 1995; Upender et al, 1997; Yoshimi et al, 2002; Hazama et al, 2005), but their presence has been attributed to neuronal uptake of antibodies secreted from B lymphocytes (Yoshimi et al, 2002; Hazama et al, 2005). To exclude that the RNA detected in our TRAP experiments or ISH originated from lymphocytic transcripts, we performed ISH experiments on tissue sections taken from Rag1 knock-out mice, which lack mature B and T cells (Mombaerts et al, 1992). Expression of *Ighm* and *Ighg3* was maintained in spinal cords of Rag1 knock-out mice (Fig 2A–D). We then extended our expression analysis to supraspinal sites and detected expression of *Ighm* but not *Ighg3* at several supraspinal sites including the cerebral cortex and the hippocampus with virtually identical expression patterns in wild-type and Rag1 knock-out mice (Fig 2E and F).

To also exclude RNA cross contaminations between different neuronal cell populations, we used multiplex ISH and investigated the expression of *Ighm* and *Ighg3* in different populations of spinal interneurons. These experiments indicated a high degree of co-expression of *Ighg3* with *Gad67*, a marker of inhibitory neurons (Fig 3A). Quantification of this co-localization demonstrated that almost all *Ighg3* expressing cells were *Gad67* positive (98% ± 6%) (Figs 1B and 3A and D), confirming the results of the TRAP experiments (Fig 1A and Table S1). Conversely, we found that *Ighm* was expressed both by inhibitory (vGAT[+]; *Ighm*[+] = 36.9% ± 10.6% of all *Ighm*[+]) (Fig 3B and D) and excitatory (vGluT2[+], *Ighm*[+] = 44.2% ± 9.9% of all *Ighm*[+]) (Fig 3C and D) spinal neurons at roughly equal levels, which is again in line with the TRAP data (Fig 1A and Table S1).

### *Ighm* expression in neurons is initiated at a hitherto unidentified 5′ exon

When analyzing the translatomes of inhibitory and excitatory spinal neurons, we found no evidence for the expression of variable regions (V, D, and J segments) in neurons (Table S1). As the transcription of regular immunoglobulin heavy chains (in B lymphocytes) is initiated at the first exon of the V segment, this finding suggested that transcription in neurons must be initiated at an alternative site. To identify such an alternative transcription start site, we performed 5′ RACE (Rapid Amplification of CDNA Ends) on reverse transcribed mRNA isolated via the TRAP approach from spinal interneurons. We used reverse nested primers located at the 5′ start of exon 2 of the immunoglobulin heavy chain genes (Fig 4A). Exon 2 is contained in both of the known splice variants of either the *Ighm* or *Ighg3* gene. Most likely, because of the very low expression of *Ighg3*, we were not able to obtain a 5′ RACE amplicon for *Ighg3*. However, a single amplicon of about 450 bp was obtained for *Ighm* (Fig 4B). We therefore focused our

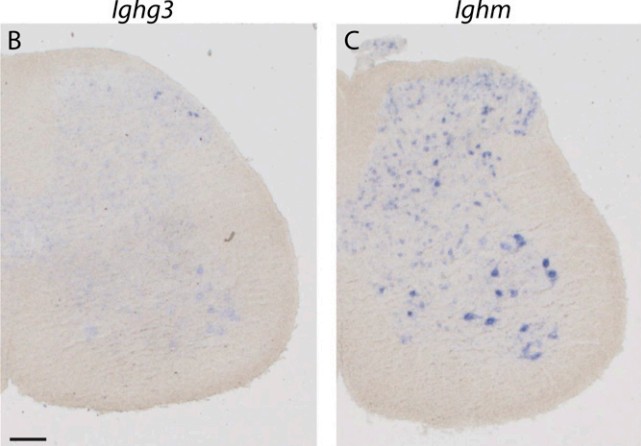

**Figure 1. Spinal expression of *Ighg3* and *Ighm*.**
**(A)** Expression of *Ighm* and *Ighg3* depicted as normalized read count. Read counts were determined in RNA samples extracted from excitatory (vGluT2+), inhibitory (vGAT+), and a subset of inhibitory (Gad67+) spinal neurons after performing RNA sequencing (Das Gupta et al, 2021). Also depicted are differences in expression (fold change) when comparing vGAT versus vGluT2 and Gad67 versus vGluT2 samples. **(B, C)** Expression pattern analysis of *Ighg3* (B) and *Ighm* (C) in adult spinal cord sections of the mouse using in situ hybridization. Scale bar: 100 μm.

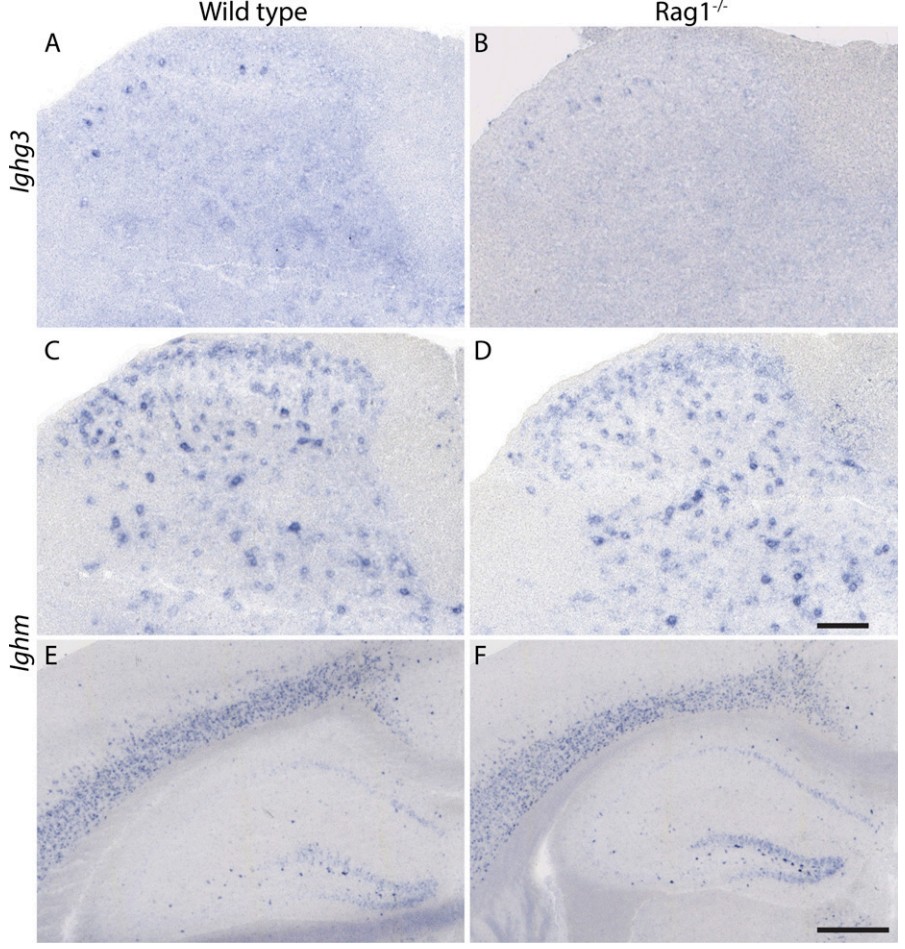

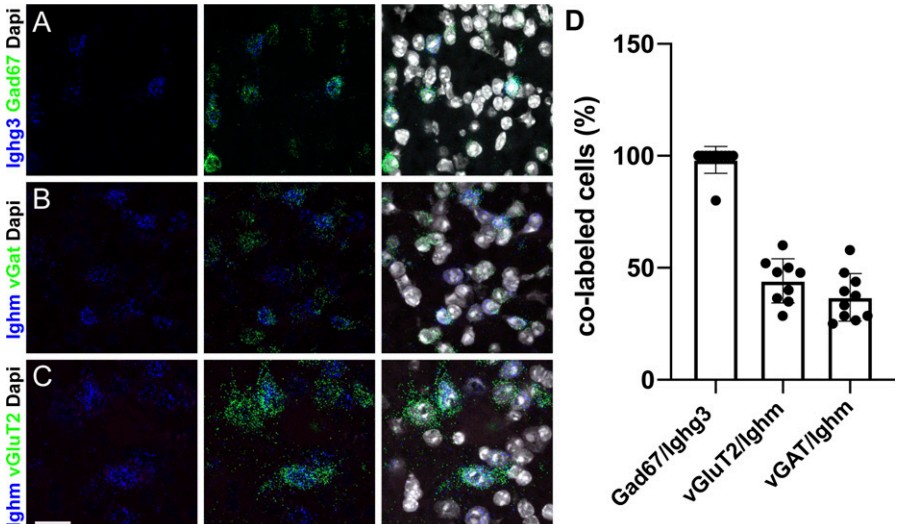

**Figure 2.  Expression of *Ighg3* and *Ighm* persists in the CNS and Rag1$^{-/-}$ mice.**

**(A, B)** In situ hybridization using an antisense probe directed against *Ighg3* on spinal cord sections of adult wild-type (A) and Rag1$^{-/-}$ (B) mice. **(C, D, E, F)** In situ hybridization using an antisense probe directed against *Ighm* on spinal cord sections (C, D) and forebrain sections (E, F) of adult wild-type (C, E) and Rag1$^{-/-}$ (D, F) mice. **(A, B, C, D, E, F)** Scale bar: (A, B, C, D) 100 μm, (E, F) 500 μm.

subsequent analysis on *Ighm*. We subcloned and sequenced the *Ighm* specific amplicon. Genomic alignment showed that the amplicon contained the first annotated exon of the *Ighm* gene but also an additional so far unknown transcript of 91 bp located upstream of the annotated exon 1. When aligning this sequence to a reference genome (ENSEMBL *Mus musculus* version 100.38 [GRCm38.p6] chromosome 12: 113,418,039–113,424,940 [Yates et al, 2019]), we found that it located ~1.2 kb upstream of the annotated exon 1 and

**Figure 3.  Expression of *Ighg3* and *Ighm* can be detected in subtypes of spinal neurons.**

**(A, D)** Multiplex in situ hybridization indicates that *Ighg3* (blue) expression occurs only in a subset of *Gad67*$^+$ inhibitory interneurons (green) (A, D). **(B, C, D)** Multiplex in situ hybridization using probes against *Ighm* (blue), vGAT (green) (B), and vGluT2 (green) (C) demonstrates expression of *Ighm* in excitatory and inhibitory spinal neurons (B, C, D). **(D)** Quantification of the number of *Ighg3*/*Ighm*–positive cells that also express the indicated marker. Error bars: SD. Sale bar: 20 μm.

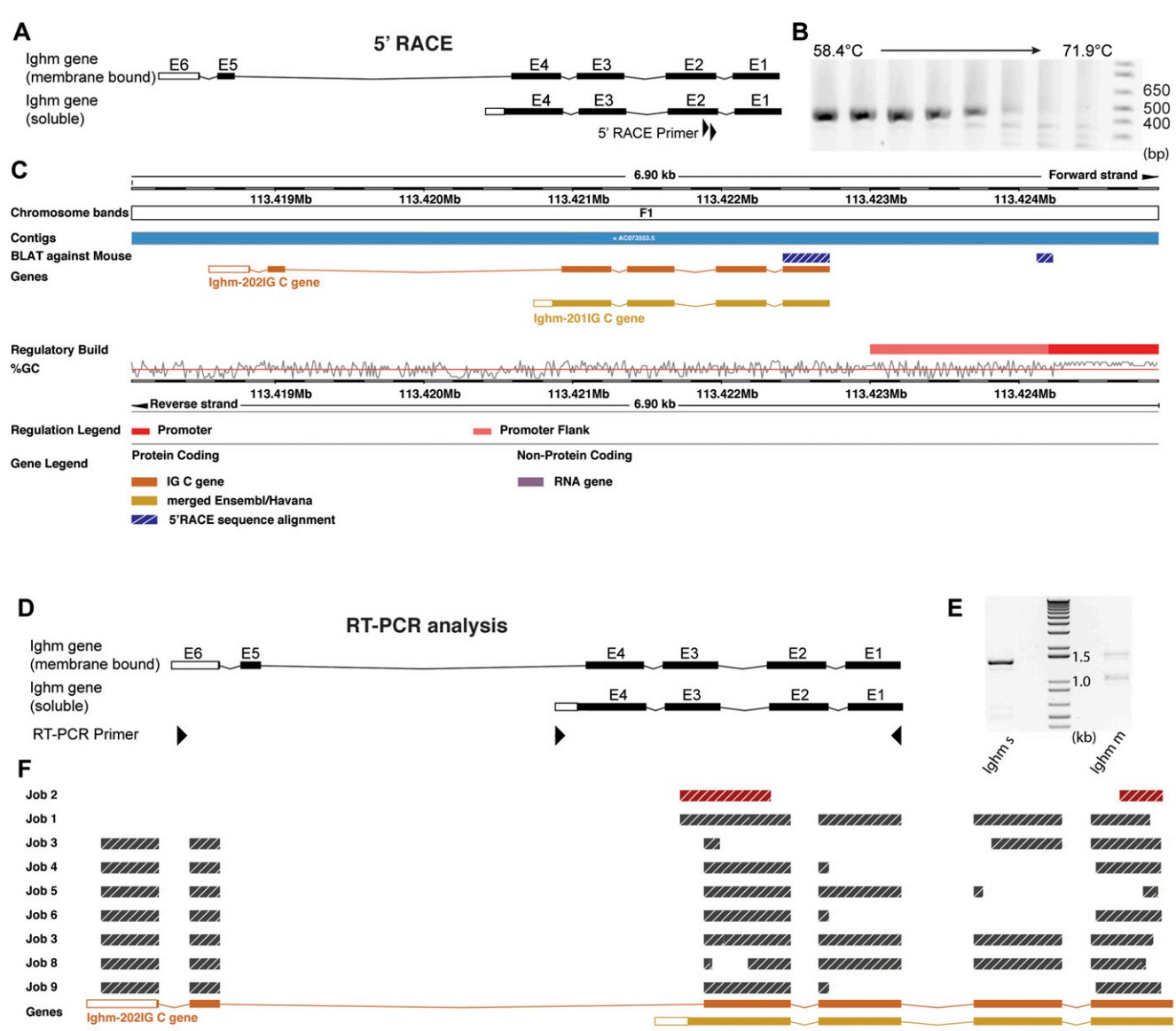

**Figure 4. Analysis of the *Ighm* transcript that is produced in spinal neurons.**
**(A, B, C)** 5' Rapid Amplification of CDNA Ends (RACE) assay for *Ighm*. **(A)** Schematic illustration of the Exon structure of the two annotated *Ighm* transcript variants. The gene specific 5' RACE primers are indicated by black triangles. **(B)** PCR results using the second gene specific primer for amplification. Depicted are PCR fragments produced by the indicated temperature gradient. **(C)** Alignment of the sequenced RACE amplicon against a reference genome (Ensembl *M. musculus* version 100.38 [GRCm38.p6] chromosome 12: 113,418,039–113,424,940 [Yates et al, 2019]). The RACE amplicon is indicated in hatched blue and white. In addition, indicated are annotated exons of the two transcript variants of *Ighm*, promoter-flanking regions (light red), and putative promoter sequences (red) as well the genomic location and the GC content of the genomic sequence. Note that the 5' sequence of the RACE amplicon lies outside of the annotated *Ighm* gene segment and next to a putative promoter sequence. **(D, E, F)** RT–PCR analysis of *Ighm* transcripts produced in spinal neurons. **(D)** Schematic illustration of the Exon structure of the two annotated *Ighm* transcript variants. The location of the used RT–PCR primers is indicated by black triangles. **(E)** RT–PCR using primers specific for the transcript variant encoding the soluble form of *Ighm* (Ighm s) produced one dominant band. RT–PCR using primers specific for the transcript variant encoding the membrane bound form of *Ighm* (Ighm m) produced two visible bands. **(F)** Alignment of sequenced RT–PCR amplicons against a reference genome (Ensembl *M. musculus* version 100.38 [GRCm38.p6] chromosome 12: 113,418,039–113,424,940 [Yates et al, 2019]). Indicated are the sequences of the various RT–PCR amplicons and the annotated exons of *Ighm*-s (Ighm-201Ig C gene) and *Ighm*-m (Ighm-202Ig C gene).

directly adjacent to a sequence that has been annotated in the ENSEMBL regulatory build (Zerbino et al, 2015) as a promoter (Fig 4C).

These results are thus in line with our RNAseq data, suggesting the absence of V, D, and J elements in the spinal neuronal transcript of *Ighm* and suggest that the neuronal transcript of *Ighm* (Fig S1) is initiated from a predicted promoter sequence 1.2 kb upstream of the first annotated exon and starts with a hitherto unidentified exon.

## Alternative splicing of *Ighm* in neurons

In B lymphocytes, IgM can be expressed in a membrane bound, receptor-like form, or as a secreted protein. The choice between the two isoforms depends on alternative splicing where alternative 3' ends of Exon 4 of *Ighm* are used. We performed RT–PCR on mRNA, isolated from excitatory and inhibitory neurons via the TRAP

approach, to investigate which of these isoforms are expressed in neurons. To this end, we combined a primer located at the start of exon 1 with primers that specifically bind to the 3′ UTR of either the membrane bound or the soluble IgM variant (Fig 4D). RT–PCR reactions undertaken with the primer binding to the 3′ UTR of the soluble variant produced one dominant band at ~1,400 bp and a few weaker bands below (Fig 4E). Analogous experiments using the membrane bound form specific primer produced two bands, one at ~1,100 bp and one at 1,600 bp. Subsequent subcloning, sequencing, and alignment of these RT–PCR products to a reference genome (ENSEMBL *M. musculus* version 100.38 (GRCm38.p6) chromosome 12: 113,418,039–113,424,940 [Yates et al, 2019]) demonstrated the presence of the full-length transcripts encoding the soluble (Exon 1–4, Fig 4F–Job1) and the membrane bound (Exon 1–6, Fig 4F–Job7) variants. Some of the sequenced amplicons displayed deletions of parts of exons or entire exons (Fig 4F–Job2–6 and 8–9), suggesting the presence of alternatively spliced transcripts which may further increase the diversity of proteins produced from *Ighm* in neurons.

### Are neuronal Fc-IgM transcripts translated into protein?

In B lymphocytes, not only transcription but also translation of IgM is initiated at the exon encoding the V segment. We found no evidence for the presence of RNA containing this V segment in neurons and furthermore demonstrated that the neuronal *Ighm*

transcript starts with a novel 5′ exon. We therefore screened the neuronal *Ighm* transcript for the presence of a putative translation initiation site (translation initiation site/start ATG) using ATGpr (https://atgpr.dbcls.jp [Salamov et al, 1998; Nishikawa et al, 2000]). Several ATGs were identified. Among those, the second ATG present in the neuronal transcript (located at position 165–168 bp) (Fig S1) had the highest reliability score (44%) and complied best with the Kozak rule. This ATG lies within the reading frame of the antibody encoding transcript and should lead to the production of a 433aa soluble version and a 451aa membrane bound form of *Ighm*. We next used SMART (Schultz et al, 1998; Letunic et al, 2021), trRosetta (Yang et al, 2020) and RaptorX (Wang et al, 2017) to identify protein domains and the protein structure of the putative protein encoded by the neuronal *Ighm* transcript. SMART's domain architecture analysis tool (http://smart.embl-heidelberg.de/) predicted four Ig domains in both transcripts and an additional transmembrane domain in the transcript variant encoding a potential membrane bound form of *Ighm* (Fig 5A and B). The two protein structure analysis tools (trRosetta and RaptorX) also predicted a protein with four immunoglobulin domains characterized by two layers of β-pleated sheets composed of strands of antiparallel polypeptide chains (Fig 5C and membrane bound Fc-IgM Video 1 and soluble Fc-IgM Video 2).

To obtain additional support for *Ighm* derived proteins in neurons, we conducted immunohistochemistry and Western blot

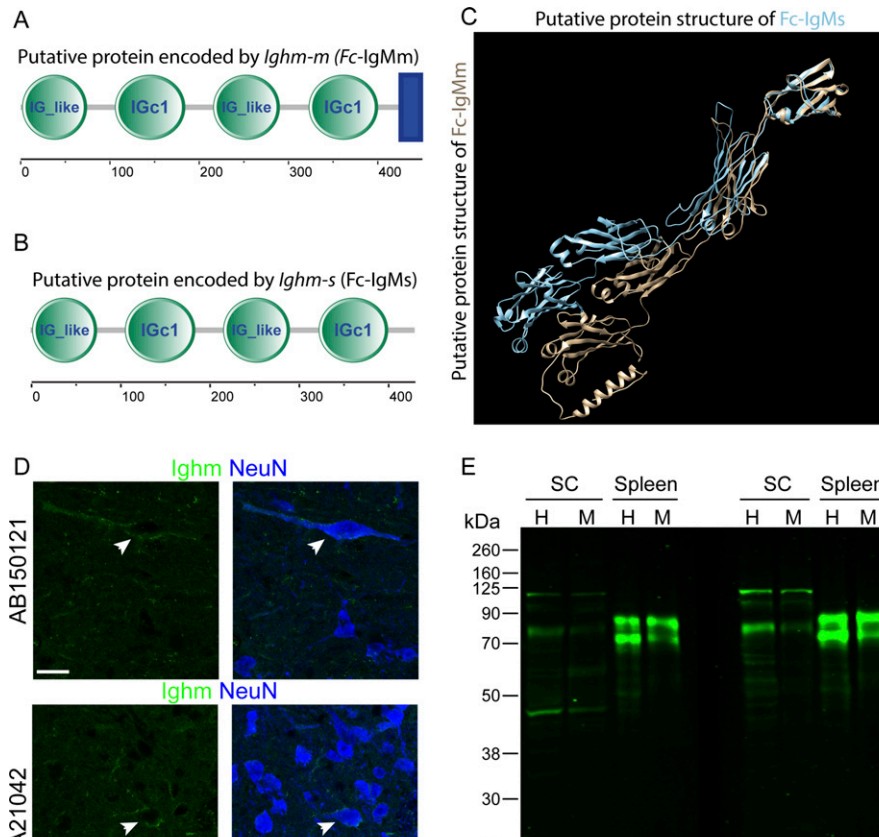

**Figure 5. Expression of a Fc-IgM protein in neurons.**
**(A)** Domain structure analysis using SMART's domain architecture analysis tool (http://smart.embl-heidelberg.de/) of the membrane bound form encoded by the neuronal *Ighm* predicts 4 Ig domains (IG_like and IGc1) and one transmembrane domain (blue rectangle). **(B)** Domain structure analysis of the soluble bound form encoded by the neuronal *Ighm* predicts the same four Ig domains (IG_like and IGc1) but no transmembrane domain. **(C)** Two protein structure analysis tools (trRosetta and RaptorX) have been used to predict the structure of the putative Fc-IgM proteins. An overlay of the membrane bound form (brown) and soluble form (blue) of Fc-IgM is depicted. **(D)** Immunohistochemical analysis of spinal cord sections using to different antibodies (A21042 and AB150121) directed against the Fc-domain of IgM. Co-labeling of IgM (green) and NeuN is depicted. **(E)** Western blot analysis of spinal cord tissue and spleen tissue homogenates using two different antibodies (Ab9167 and M8644). The homogenized tissue was either analyzed as total extract (H) or after an additional centrifugation as membrane fraction (M). Molecular weight markers are indicated on the left side in kD. Scale bar: 20 $\mu m$.

analyses of spinal tissue, using four different commercially available anti-mouse-IgM antibodies. In immunohistochemistry experiments, two different antibodies produced signals in the spinal gray matter, where they were located around individual neuronal somata (Fig 5D). For Western blot analysis, we extracted spinal cord tissue from adult mice after ACSF perfusion. The analysis was performed on total extracts and membrane preparations of spinal cord and, as a positive control, spleen tissue. Using two different antibodies, we detected two prominent bands in the spleen samples (Fig 5E). Neither of these two bands were detected in the samples taken from the spinal cord, indicating that the ACSF perfusion had successfully removed lymphocytes from the tissue. However, both antibodies also detected a prominent band in the spinal cord samples running below the higher molecular weight band detected in the spleen samples (Fig 5E), consistent with the lower molecular weight of the predicted neuronal Fc-IgM protein lacking the variable domain. Additional higher and lower molecular weight bands were detected, which may represent either nonspecific antibody reactions, degradation products of IgM or neuronal Fc-IgM or protein products resulting from alternative splicing or posttranslational modification (Arnold et al, 2007).

Together with our initial observation, the detection of *Ighm* and *Ighg3* transcripts bound by translating ribosomes of spinal neurons, immunohistochemical and Western blot analyses suggest that neuronal *Ighm* transcripts are translated into proteins. We therefore propose a functional role of IgM and IgG3 derived constant domain proteins in the CNS of the mouse.

# Discussion

### Potential functions of immunoglobulin heavy chain gene products in neurons

We are well aware of the provocative nature of our findings. However, there is previous evidence for an immune reaction independent signaling role of Ig in the healthy CNS. Some reports hint at a potential function of the IgM constant domain in oligodendrogenesis. Oligodendrocyte precursors express the IgM receptor Fcα/μR (encoded by the *Fcmar* gene) (Nakahara et al, 2003). Its neutralization inhibits oligodendrocyte precursor proliferation and reduces the proportion of myelinated neurons in the early postnatal CNS. Other experiments from the same group suggest that the activation of these Fcα/μR receptors occurs via IgM secreted by a subtype of B lymphocytes (Tanabe & Yamashita, 2018). Our present study indicates that such activation may also occur via Igs secreted from neurons. The *Ighm* transcripts identified in the present study encode the Ig Fc region that binds to the Fcα/μR. Because part of the *Ighm* transcripts encode for the soluble isoform of the IgM constant heavy chain domain, the proteins translated from these transcripts may serve as diffusible signals for neuron to oligodendrocyte precursor cell communication.

Neuron-derived membrane bound Fc-IgM or Fc-IgG3 might also function as cell adhesion molecules similar to other members of the Ig superfamily expressed in the nervous system. A significant portion of the Ig superfamily members is present on the surface of cells of the nervous system. Here, they often act as cell adhesion molecules mediating the interaction in-between cells or the interaction of cells with the extracellular matrix. As such Ig super-family proteins are required for several steps during neuronal development, for example, neuronal migration, pathfinding, target recognition, and synapse formation as well as for circuit maintenance and reorganization in the adult (Rougon & Hobert, 2003; Stoeckli, 2004; Sytnyk et al, 2017; Zinn & Ozkan, 2017; Sanes & Zipursky, 2020).

Neuronal IgM constant heavy chain domain proteins may also act as activators of the complement system, which has been implicated in synaptic development and maintenance (Stephan et al, 2012; Presumey et al, 2017). The complement system has recently been described to be involved in the recruitment of microglia to synaptic terminals in the developing nervous system and to potentially play a role in neurodegenerative disorders. It has been suggested that excessive synaptic connections, which are built during development, become tagged, for example, by complement components such as C1q, and are thus primed for elimination by microglia in a process termed synaptic pruning (Stephan et al, 2012; Sakai, 2020). IgM can activate the complement via its constant region which binds to C1q (Sharp et al, 2019). Neuronally expressed Fc-IgM may therefore be involved in attracting and activating microglia.

### Conclusions

It has meanwhile become apparent that expression of Ig-superfamily proteins in the immune system or in the nervous system is not mutually exclusive. In fact, a number of proteins that have initially been identified in the immune system have later on also been detected in the nervous system and vice versa (Boulanger, 2009). A prominent example in invertebrates is Down syndrome cell adhesion molecule proteins (Schmucker et al, 2000; Watson et al, 2005). In vertebrates, key players of the adaptive immune system, such as members of the major histocompatibility complex class I (MHCI) family and MHCI-binding immunoreceptors, are also implicated in synapse formation and maintenance (Boulanger, 2009). However, immunoglobulins themselves are still widely believed to be exclusively expressed by B cells. Our study has uncovered that two genes (*Ighm* and *Ighg3*), encoding eponyms of the Ig-superfamily, are expressed in neurons of the spinal cord and at supraspinal sites with potential functions in CNS development or maintenance.

# Materials and Methods

### Animals

Experiments were performed on 6- to 10-wk-old mice kept at a 12:12 h light/dark cycle that received food and water ad libitum and were kept in a 12 h light/dark cycle. All methods were carried out in accordance with relevant guidelines and regulations. All animal experiments have been applied for. They have been evaluated by the Commission on Animal Experimentation of the canton of Zurich.

**Bright-field imaging of ISHs and fluorescent imaging of FISHs and IHCs were performed using the following microscopes:**

| Microscope | Objectives | Software | Purpose |
|---|---|---|---|
| Zeiss Axio Scan.Z1 slidescanner | 5×/0.25 Fluar air | ZEN 2 slidescan (blue edition) | Bright field images of chromogenic ISHs |
| | 10×/0.45 Plan-Apochromat air | | |
| Zeiss LSM 800 with Airyscan/confocal | 25×/0.8 Plan-Neofluar oil | ZEN 2.6 (blue edition) | Overview and analysis images of multiplex FISH |
| | 40×/1.4 Plan-Apochromat oil | | |

The Commission recommended the cantonal veterinary office (Kanton Zürich, Gesundheitsdirektion, Veterinäramt, Zollstrasse 20, CH-8090 Zürich) to accept the respective applications (license ZH011/2019). This license covers all experiments described in this study. Animal experiments were conducted adhering to the ARRIVE guidelines.

## Immunohistochemistry (IHC)

After injection of an overdose of pentobarbital and the subsequent loss of pinch reflexes, mice were transcardially perfused with ~20 ml of ice-cold artificial cerebrospinal fluid (ACSF, pH 7.4) or PBS (pH 7.4), followed by 100 ml of 4% ice-cold PFA (PFA, in PBS or 0.1 M sodium phosphate buffer [PB], pH 7.4). The lumbar spinal cord was dissected and post-fixed for 1.5–2 h in 4% PFA solution (in PBS or 0.1 M PB, pH 7.4), followed by incubation in 30% sucrose (in PBS or 0.1 M PB, pH 7.4) for cryoprotection at 4°C overnight. Cryoprotected spinal cords were embedded in NEG50 frozen section medium (Richard-Allen Scientific) and stored at −80°C until cutting into 25 $\mu$m thick sections on a Hyrax C60 Cryostat (Carl Zeiss). Sections were mounted on Superfrost Plus glass slides (Thermo Fisher Scientific) and stored at −80°C.

For immunofluorescence staining, the slides were washed for 5 min in PBS to remove the embedding medium, followed by blocking with 5% normal donkey serum (RRID:SCR_008898; AbD Serotec) in 0.1% Triton X-100–PBS for at least 30 min at RT. Sections were incubated with primary antibodies (see Resource table) in the blocking solution at 4°C overnight, followed by three washes in PBS for 5 min and incubation with secondary antibodies (fluorophore-coupled donkey antibodies; Jackson ImmunoResearch) in blocking solution at RT for 30–60 min. Subsequently, sections were washed in PBS. Finally, sections were covered with DAKO fluorescent mounting medium (RRID:SCR_013530; Dako) and coverslips.

## ISH

For ISHs, 6–10 wk old, naïve male C57BL/6J mice were used. For spinal cord and brain preparation, the spinal cord and brain were dissected, immediately after decapitation of the mouse. After dissection, spinal cords were snap frozen in liquid nitrogen and stored at −80°C until embedding in NEG50 frozen section medium (Richard-Allen Scientific). Frozen blocks were again stored at −80°C until cutting into 16–25 $\mu$m thick sections and mounted as described for IHC.

ISHs were performed with DIG-labeled riboprobes. DNA for *Ighm* and *Ighg3* was amplified with gene-specific sets of PCR primers from cDNA templates generated from RNA isolated from mouse spinal cords. The PCR fragments were cloned into Teasy vector

(Promega), and sequence was verified before use for riboprobe generation. ISH were performed as previously described (Wildner et al, 2008, 2013).

For multiplex FISH, the manual RNAscope Multiplex Fluorescent Assay (ACD, Cat. no. 320850; Bio-Techne) was used. The manufacturer's pretreatment protocol for fresh frozen tissue (document no. 320513, rev. date 11052015) and detection protocol (document no. 320293-UM, rev. date 03142017) were followed. The fluorophore alternatives (Amp 4 Alt) were chosen in such a way that when possible, the weakest expressing gene would lie in red channel (Atto 550) and not in the far-red channel (Atto 647). The 3-plex negative control probe was amplified with the corresponding Amp 4 Alt. Probes are listed in the resource table.

## Image acquisition and analysis

For fluorescent imaging, the pinhole was set to 1 airy unit for every channel, which were scanned sequentially to avoid overlapping emission spectra or with a combination of the ultraviolet and infrared channel in one track, where emission spectra overlap is minimal.

For the analysis, the cell counter plugin of ImageJ was used. Three hemi-sections from three animals were analyzed. Ratios were calculated per animal and then averaged.

## 5′ RACE and RT–PCR

5′ RACE was carried out on neuron specific cDNA that was generated using the TRAP approach (Das Gupta et al, 2021). 5′ RACE was conducted using the FirstChoice RLM-RACE Kit (Cat. no. AM1700; Thermo Fisher Scientific) according to the manufacturer's instructions. *Ighm* specific amplicons were generated with the following nested primer pairs: first PCR, o1108_5′ RACE_Outer (GCTGATGGCGATGAATGAACACTG) and o1106_4_Reverse primer (GAGTTTAGACTTGCGTGGTG). Second (nested) PCR, o1115_5′ RACE_Inner (ACTGCGTTTGCTGGCTTTGATG) and o1107_5_Reverse primer (GTGGTGGGACGAACACATTT).

RT–PCR was carried out on neuron specific cDNA (Das Gupta et al, 2021). To increase specificity of the PCR we performed two rounds of PCRs using the following nested primers: first PCR-Ighm-m; n885a_Ighm1_s (CTGACATGGTTAGTTTGCATACACAGAG) and n886a_Ighm1_2_a (AGTCAGTCCTTCCCAAATGTCTTCC). Second PCR-Ighm-m; o885a_Ighm1_s (AGGGCCTGCCTGGTTGAG) & o886a_Ighm1_2_a (CTCGTCTCCTGCGAGAGCC). First PCR-Ighm-s; n884_Ighm2_s (CCATGTGACATTTGTTTACAGCTCAG) and n886a_Ighm1_2_a (AGTCAGTCCTTCCCAAATGTCTTCC). Second PCR-Ighm-s; o884_Ighm2_s (GTCTGTGGGCCAGACATTGC) and o886a_Ighm1_2_a (CTCGTCTCCTGCGAGAGCC).

**Resource table.**

| Reagent | Resource | Identifier |
|---|---|---|
| Antibodies | | |
| Goat anti-IgM | AB9167 (Abcam) | RRID:AB_956044 |
| Goat anti-IgM | M8644 (Sigma-Aldrich) | RRID:AB_260700 |
| Goat anti-IgM | AB150121 (Abcam) | RRID:AB_2801490 |
| Goat anti-IgM | A21042 (Thermo Fisher Scientific) | RRID:AB_2535711 |
| Guinea pig anti-NeuN | 266004 (SynapticSystems) | RRID:AB_2619988 |
| RNAscope multiplex FISH probes | | |
| Mm-GAD1 (Gad67) | ACD; Bio-Techne | 400951 |
| Mm-Slc32a1 (vGat) | ACD; Bio-Techne | 319191 |
| Mm-Slc17a6 (vGuT2) | ACD; Bio-Techne | 319171 |
| Mm-Ighm | ACD; Bio-Techne | 514621 |
| Mm-Ighg3 | ACD; Bio-Techne | 514611 |
| Positive control probe Mm-Ppib | ACD; Bio-Techne | 313911 |
| Negative control probe-DapB | ACD; Bio-Techne | 310043 |
| 3-plex negative control probe | ACD; Bio-Techne | 320871 |

## Western blot analysis

Spinal cord and spleen tissue were rapidly dissected and immediately frozen on dry ice and stored at –80°C until used. To avoid contamination of spinal cord samples with blood, the mice were transcardially perfused with ACFS before dissection. For Western blot analysis, the tissue was thawed and homogenized in 10 volumes of PBS containing the protease inhibitor cocktail CompleteMini (Roche Diagnostics). The homogenate was centrifuged for 10 min at 800$g$ and the supernatant was recovered. One portion of the supernatant was stored on ice (total extract), whereas the remaining supernatant was centrifuged at 60,000$g$ for 20 min. The pellet (membrane fraction) was resuspended by sonication. After protein determination using the Bradford protein assay (Bio-Rad), the samples were incubated with Laemmli sample buffer (Bio-Rad) for 60 min at 37°C. Protein aliquots containing 20 $\mu$g were resolved, along with 5 $\mu$l pre-stained protein ladder (Chameleon Duo, 928-60000; LI-COR Biosciences), by SDS–PAGE using 10% mini-gels (Mini Protean 3; Bio-Rad). Proteins were transferred onto nitrocellulose membranes in a Trans-Blot Semi-Dry Transfer Cell (Bio-Rad) at 15 V for 90 min using 39 mM glycine, 48 mM Tris, 1.3 mM SDS, and 20% methanol as transfer buffer. The blots were blocked for 1 h with 5% non-fat dry milk in PBS. Incubation with primary antibody was done at 4°C overnight with anti-IgM antibodies (1:200, Cat. no. M8644; Sigma-Aldrich and Cat. no. ab9167; Abcam) diluted in TBST (10 mM Tris, pH 7.4, 150 mM NaCl, and 0.05% Tween 20) containing 5% non-fat dry milk. The blots were then washed five times for 5 min with TBST and incubated with donkey anti-goat IRDye 800CW conjugated secondary antibody (1:8,000; LI-COR Biosciences) for 1 h at room temperature. After washing in TBST and finally TBS, immunoreactivity was detected using the Odyssey CLx imager (LI-COR).

## Supplementary Information

## Acknowledgements

We are grateful to Adriano Aguzzi for providing Rag1 knock-out mice. This work was supported by a Swiss National Science Foundation (SNSF) SPARK grant (CRSK-3_190622) to H Wildner and an ETHIIRA grant (0-20157-15) from ETH Zurich to HU Zeilhofer.

## Author Contributions

L Scheurer: data curation, investigation, visualization, and methodology.
RR Das Gupta: data curation, investigation, visualization, and methodology.
A Saebisch: data curation, validation, and methodology.
T Grampp: data curation and methodology.
D Benke: investigation and methodology.
HU Zeilhofer: conceptualization, funding acquisition, and writing—original draft.
H Wildner: conceptualization, supervision, funding acquisition, investigation, visualization, methodology, and writing—original draft, review, and editing.

## Conflict of Interest Statement

The authors declare that they have no conflict of interest.

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
