## [Reviewer comments · Life Science Alliance]

Life Science Alliance

Expression of Immunoglobulin constant domain genes in neurons of the mouse central nervous system

Hendrik Wildner, Louis Scheurer, Rebecca Das Gupta, Annika Saebisch, Thomas Grampp, Dietmar Benke, and Hanns Zeilhofer

DOI: <https://doi.org/10.26508/lsa.202101154>

Corresponding author(s): Hendrik Wildner, University of Zurich and Hanns Zeilhofer, Institute of Pharmacology and Toxicology

Review Timeline:

Submission Date:	2021-07-08
Editorial Decision:	2021-07-23
Revision Received:	2021-08-06
Editorial Decision:	2021-08-09
Revision Received:	2021-08-16
Accepted:	2021-08-17

Transaction Report:

July 23, 2021

Re: Life Science Alliance manuscript #LSA-2021-01154-T

Dr. Hendrik Wildner
University of Zurich
Institute of Pharmacology and Toxicology
Winterthurerstrasse 190
Zurich, Zurich CH-8057
Switzerland

Dear Dr. Wildner,

Thank you for submitting your manuscript entitled "Expression of Immunoglobulin constant domain genes in neurons of the mouse central nervous system" to Life Science Alliance. The manuscript was assessed by an expert reviewer, whose comments are appended to this letter. We invite you to submit a revised manuscript addressing the Reviewer comments.

When submitting the revision, please include a letter addressing the reviewer's comments point by point.

Thank you for this interesting contribution to Life Science Alliance. We are looking forward to receiving your revised manuscript.

Sincerely,

- A letter addressing the reviewer's comments point by point.
- An editable version of the final text (.DOC or .DOCX) is needed for copyediting (no PDFs).
- High-resolution figure, supplementary figure and video files uploaded as individual files: See our detailed guidelines for preparing your production-ready images, <https://www.life-science-alliance.org/authors>
- Summary blurb (enter in submission system): A short text summarizing in a single sentence the study (max. 200 characters including spaces). This text is used in conjunction with the titles of papers, hence should be informative and complementary to the title and running title. It should describe the context and significance of the findings for a general readership; it should be written in the present tense and refer to the work in the third person. Author names should not be mentioned.

B. MANUSCRIPT ORGANIZATION AND FORMATTING:

Reviewer #1 (Comments to the Authors (Required)):

In this study the authors provide several independent lines of evidence to show that two different immunoglobulin heavy chains are expressed by neurons in the mouse central nervous system. This is clearly a very unexpected finding, as expression of immunoglobulins has been assumed to be restricted to lymphocytes, and I have no doubt that it will be of considerable interest to many neuroscientists. Although the authors are not yet able to explain the functional significance of their findings, they discuss several possible roles for the immunoglobulin heavy chains. These include involvement in oligodendrocyte development, neuronal migration and pathfinding, and the generation and pruning of synapses. The work is of high quality, the findings are convincing and the paper is very well written.

I have only a few minor suggestions for improvement.

P2: it would help to say something about the possible significance of this new finding at the end of

the Introduction.

P3: the distribution of Ighg3 is interesting, as it is restricted to inhibitory interneurons and shows a very restricted laminar pattern. Can the authors say what proportion of inhibitory neurons in the superficial dorsal horn were Ighg3-positive? Also, can they relate this to any of the neurochemical populations that have been identified in this region?

P4: two typos - "lymphocytes" in line 9; "characterised" in the 6th last line.

P5: the last sentence of the 2nd paragraph could be improved. I'm not sure that you can propose an "unknown functional role".

P6 3rd last line: "eponyms"?

P7: the "Animals" section needs to be tidied up to avoid repetition in the first sentence and clarify the ethical permission.

P8 line 4: "neuropeptide combination" is probably carried over from a previous paper.

Fig 1 legend: "two types of inhibitory . . . spinal neurons". This is not quite correct, as there will be extensive overlap between cells defined by expression of VGAT and GAD67

Fig 4 legend: the last two words (Note amplicons need to be removed or explained).

Fig 5 legend line 9: brown would be better than green to describe the membrane bound form. There is also some red staining in the top two images in part D, which should be removed.

We would like to thank the reviewer for acknowledging the novelty of our findings and his comments. Please find below our point-by-point reply.

Reviewer #1 (Comments to the Authors (Required)):

In this study the authors provide several independent lines of evidence to show that two different immunoglobulin heavy chains are expressed by neurons in the mouse central nervous system. This is clearly a very unexpected finding, as expression of immunoglobulins has been assumed to be restricted to lymphocytes, and I have no doubt that it will be of considerable interest to many neuroscientists. Although the authors are not yet able to explain the functional significance of their findings, they discuss several possible roles for the immunoglobulin heavy chains. These include involvement in oligodendrocyte development, neuronal migration and pathfinding, and the generation and pruning of synapses. The work is of high quality, the findings are convincing and the paper is very well written.

I have only a few minor suggestions for improvement.

P2: it would help to say something about the possible significance of this new finding at the end of the Introduction.

We have added a sentence at the end of the introduction.

P3: the distribution of Ighg3 is interesting, as it is restricted to inhibitory interneurons and shows a very restricted laminar pattern. Can the authors say what proportion of inhibitory neurons in the superficial dorsal horn were Ighg3-positive? Also, can they relate this to any of the neurochemical populations that have been identified in this region?

We agree with the reviewer that the expression pattern of Ighg3 is interesting. We have not yet performed in depth co-expression analysis to determine which of the 15 different types of inhibitory spinal interneurons are co-expressing Ighg3. However, we are planning to follow up on the function of Ighg3 and Ighm including additional in-depth co-expression analysis in future studies.

P4: two typos - "lymphocytes" in line 9; "characterised" in the 6th last line.

Has been corrected.

P5: the last sentence of the 2nd paragraph could be improved. I'm not sure that you can propose an "unknown functional role".

Has been modified.

P6 3rd last line: "eponyms"?

We wanted to highlight that, while many different members of the Ig-superfamily are expressed in the nervous system, there is no reference yet of an expression of the name giving members (immunoglobulins) of the Ig-superfamily in neurons. While none of the authors is a native speaker, we thought that the word "eponym" would characterize something or someone, in this case immunoglobulins, as the first of which others are named after. If the Editors or the reviewer feel that this is an incorrect use of the word, we can change it.

P7: the "Animals" section needs to be tidied up to avoid repetition in the first sentence and clarify the ethical permission.

We have tidied up the animal section.

P8 line 4: "neuropeptide combination" is probably carried over from a previous paper.

Has been deleted.

Fig 1 legend: "two types of inhibitory . . . spinal neurons". This is not quite correct, as there will be extensive overlap between cells defined by expression of VGAT and GAD67

We agree with the reviewer that this quote is not entirely correct and have modified it accordingly.

Fig 4 legend: the last two words (Note amplicons need to be removed or explained).

Have been removed.

Fig 5 legend line 9: brown would be better than green to describe the membrane bound form. There is also some red staining in the top two images in part D, which should be removed.

The red staining has been removed and we changed the description of the membrane bound form to brown.

August 9, 2021

RE: Life Science Alliance Manuscript #LSA-2021-01154-TR

Dr. Hendrik Wildner
University of Zurich
Institute of Pharmacology and Toxicology
Winterthurerstrasse 190
Zurich, Zurich CH-8057
Switzerland

Dear Dr. Wildner,

Thank you for submitting your revised manuscript entitled "Expression of Immunoglobulin constant domain genes in neurons of the mouse central nervous system". We would be happy to publish your paper in Life Science Alliance pending final revisions necessary to meet our formatting guidelines.

- please consult our manuscript preparation guidelines <https://www.life-science-alliance.org/manuscript-prep> and make sure your manuscript sections are in the correct order
- please separate the Results and Discussion section into two - 1. Results 2. Discussion, as per our formatting requirements
- please add the Twitter handle of your host institute/organization as well as your own or one of the authors in our system
- LSA allows supplementary figures, but no EV Figures; please update your callouts for the Supplementary Figure/Table in the manuscript Fig EV1A=Fig S1A/Table S1
- we encourage you to revise the figure legends for figure 5 such that the figure panels are introduced in an alphabetical order
- please add a callout for Figures 3B and C to your main manuscript text
- please add back the details that were removed from the animal approval section

Figure checks:

- missing scale bars for Figures 1B, C

LSA now encourages authors to provide a 30-60 second video where the study is briefly explained. We will use these videos on social media to promote the published paper and the presenting author. Corresponding or first-authors are welcome to submit the video. Please submit only one video per manuscript. The video can be emailed to contact@life-science-alliance.org

A. FINAL FILES:

B. MANUSCRIPT ORGANIZATION AND FORMATTING:

Sincerely,

August 17, 2021

RE: Life Science Alliance Manuscript #LSA-2021-01154-TRR

Dr. Hendrik Wildner
University of Zurich
Institute of Pharmacology and Toxicology
Winterthurerstrasse 190
Zurich, Zurich CH-8057
Switzerland

Dear Dr. Wildner,

Thank you for submitting your Research Article entitled "Expression of Immunoglobulin constant domain genes in neurons of the mouse central nervous system". It is a pleasure to let you know that your manuscript is now accepted for publication in Life Science Alliance. Congratulations on this interesting work.

DISTRIBUTION OF MATERIALS:

Again, congratulations on a very nice paper. I hope you found the review process to be constructive and are pleased with how the manuscript was handled editorially. We look forward to future exciting submissions from your lab.

Sincerely,
